# AMPs as Host-Directed Immunomodulatory Agents against Skin Infections Caused by Opportunistic Bacterial Pathogens

**DOI:** 10.3390/antibiotics13050439

**Published:** 2024-05-13

**Authors:** Subhasree Saha, Devashish Barik, Debabrata Biswas

**Affiliations:** Institute of Life Sciences, NALCO Square, Bhubaneswar 751023, Odisha, India; ssaha@houstonmethodist.org (S.S.); devashish@ils.res.in (D.B.)

**Keywords:** antimicrobial peptides, AMPs, immunomodulation, skin pathogen, immune cells, gram-positive bacteria, multi-drug resistance

## Abstract

Skin is the primary and largest protective organ of the human body. It produces a number of highly evolved arsenal of factors to counter the continuous assault of foreign materials and pathogens from the environment. One such potent factor is the repertoire of Antimicrobial Peptides (AMPs) that not only directly destroys invading pathogens, but also optimally modulate the immune functions of the body to counter the establishment and spread of infections. The canonical direct antimicrobial functions of these AMPs have been in focus for a long time to design principles for enhanced therapeutics, especially against the multi-drug resistant pathogens. However, in recent times the immunomodulatory functions performed by these peptides at sub-microbicidal concentrations have been a point of major focus in the field of host-directed therapeutics. Such strategies have the added benefit of not having the pathogens develop resistance against the immunomodulatory pathways, since the pathogens exploit these signaling pathways to obtain and survive within the host. Thus, this review summarizes the potent immunomodulatory effect of these AMPs on, specifically, the different host immune cells with the view of providing a platform of information that might help in designing studies to exploit and formulate effective host-directed adjunct therapeutic strategies that would synergies with drug regimens to counter the current diversity of drug-resistant skin opportunistic pathogens.

## 1. Introduction

Skin is the largest organ of human body that provides a general protective barrier. The skin forms the first line of defense against any external pathogen or entities. It consists of three different layers: epidermis, dermis, and hypodermis [1]. The epidermis or the epithelial layer is crucial for providing a protective layer to body against invasive pathogens, foreign entities and additionally controls loss of water and electrolytes. The different layers making up the epidermis are stratum corneum (SC), stratum granulosum (SG), stratum spinosum (SS), and stratum basale (SB) which altogether are responsible for the waterproof nature of the skin [2]. The primary cells constituting the epidermis are keratinocytes (epidermal cells), melanocytes (melanin producing cells), and Langerhans cells (immune cells). The keratinocytes present in the epidermis undergo cornification to differentiate into corneocytes. The corneocytes further form the acidic envelope also known as the cornified envelope (CE). Keratinocytes are generally tightly embedded within the CE and are mostly connected by the corneodesmosomes that form the main intercellular adhesive structures in the stratum corneum [3]. Dermis is the thick middle layer of skin. The dermis is made of two layers: the reticular dermis and the papillary dermis. Reticular dermis, the thick basal layer of dermis, is comprised of blood vessels, nerves, lymphatic system, and adipose tissues. It supports the overall movement of skin. Papillary dermis is the upper layer of dermis that contains sweat glands, hair follicles, blood vessels, adipocytes, and mostly phagocytic immune cells. The papillary dermis aids in thermoregulation of the body [4]. The hypodermis is the innermost skin layer also known as the subcutaneous layer of skin. Hypodermis aids in connecting dermal layer to muscles and bones while contributing to the insulation of the body at the same time. The hypodermis usually consists of mostly adipose and connective tissues, blood vessels, and large nerves [5]. Skin contributes to physical, chemical as well immunological protection of body against the external environment. 

From a cellular point of view, the skin consists of keratinocytes, and fibroblasts along with leucocytes like Langerhans cells, macrophages, dendritic cells, and resident memory T cells. All these together are responsible for the maintenance and protection of the skin barrier. The keratinocytes present in skin are majorly responsible for innate immune interactions during infections [6]. They express pathogen pattern recognition receptors (PRRs) like Toll-like receptors (TLRs) which get induced upon pathogen exposure. The keratinocytes are found to express TLR 1-3, 5, and 10 which produces inflammatory cytokines and chemokines, including interleukin-1*β* (IL-1*β*), IL-8, and CCL20. Upon wound infection, the TLRs trigger recruitment of local and blood leucocytes to tackle the onset of infection [7]. The Langerhans cells (LCs) also reside in the outer region of the epidermis and aid in antigen presentation to activate the immune response against any infection. Dermis region contains dendritic cells (skin origin) and macrophages which either play role of Antigen Presenting cells (APCs) and present pathogen-associated antigens to the resident T-cells or initiate immune reaction via the lymphatic system [8]. The resident T cells, also known as the Skin resident memory T (T_rm_) cells, are crucial elements of the skin-associated immune response. Apart from resident T cells, the regulatory T cells CD4^+^ FoxP3^+^ T regulatory (T_reg_) cells are also essential for regulation and maintenance of the immune response homeostasis [9]. Also, dendritic epidermal γδ T cells (DETCs) are additionally critical for recognizing the danger-associated molecular patterns (DAMPs) in case of wound and pathological conditions [10].

The skin is colonized with normal skin microbiota that is mostly commensal in nature. However, under abnormal or perturbed circumstances, these microbiota behave as opportunistic pathogens. One of the most common ways the pathogen breaches the skin barrier is via wound or tear of skin that allows entry of pathogen inside the body and starting of infection and evasion from the immediate immune response [11]. One of the best example is *Staphylococcus aureus*, which produces superantigen proteins and toxins to nullify the neutrophil attack. Staphopain A degrades elastin and leads to blockade of chemokine receptor 2 (CXCR2) [12]. Streptococci or mostly group A streptococcus (GAS) produces pneumolysin and streptolysin O (SLO) which are pore-forming in nature and aid the bacteria to evade immune cells. Apart from that, they produce c5a-peptidase and M-proteins that inactivate the complement system [13].

Among the complex series of events taking place when bacteria interacts with the skin, one of the important steps is how the resident immune cells of skin start the release of chemokines, cytokines and AMPs in order to deal with the infection. As AMPs are one of the major players in the skin related immune reactions so, in the current review we have elaborately discussed various roles of human skin AMPs including protection against the microbial infection, immunomodulation of immune cell responses, and application of AMPs in adjunct therapy against various skin infections.

In this review, we primarily focused on the non-canonical immunomodulatory functions of AMPs against Gram-positive opportunistic bacteria in skin infections, which might provide a potent avenue for shaping host-directed therapeutics against, specially, the multi-drug drug-resistant varieties of such bacteria.

## 2. Opportunistic Skin Pathogen-Associated Infections

### 2.1. Group A Streptococcal Infections

The skin has a reservoir of normal microbiota that stays in a commensal relation with the host. However, immunodeficient circumstances or breach of the skin barrier allow these commensals to become opportunistic pathogens. Streptococci are gram-positive bacteria that reside on the skin. Amongst them, *Streptococcus pneumoniae* and *Streptococcus pyogenes* (also known as group A streptococcus or GAS), are the two most critical pathogens responsible for lethal diseases [14]. Group A streptococcus causes condition ranging from mild infections like impetigo, ecthyma, cellulitis to severe life threatening ones like necrotizing fasciitis, acute glomerulonephritis and toxic shock syndrome (TSS) [15]. It is also known as the “flesh-eating” bacterium since it invades the soft tissue and destroys them during necrotizing fasciitis. There are more than 200 serotypes of group A streptococci depending on the surface protein known as M protein which is encoded by the *emm* gene [16]. GAS has several virulence factors that aids in the host pathogenesis (Figure 1). It has hyaluronic acid (HA), pili, M proteins and the *S. pyogenes* fibronectin-binding adhesin (SfbI), which allows the pathogen to adhere and colonize the nasopharynx region including tonsil epithelium and skin [17]. Apart from that, it has many pyrogenic exotoxins like exotoxins A, B, and C and superantigens like streptococcal superantigen (SSA), *Spe*A, *Spe*B, *Spe*G, *Spe*H, *Spe*J, *Sme*Z, and *Sme*Z-2 [18]. These super antigens interact with major histocompatibility complex (MHC) class II molecule and leads to the nonspecific heightened activation of T cells which further leads to the excessive production of various interleukins (IL-1, IL-6) and inflammatory cytokines such as tumor necrosis factor beta (TNFβ) and gamma interferon (IFNγ) [19].

### 2.2. Staphylococcal Infections

*Staphylococcus aureus* is the most common member of the normal skin microbiota that can turn in to an opportunistic pathogen in case of host immune compromise. Staphylococcal infections range from mild skin diseases, like impetigo, folliculitis, furunculosis, abscesses, to severe conditions, such as endocarditis, pneumonia, sepsis and toxic shock syndrome [21]. They are notoriously known to consist of highly antibiotic resistance strains like methicillin-resistant *S. aureus* (MRSA) and vancomycin-resistant *S. aureus* (VRSA) that make treatment processes highly difficult [22]. *S. aureus* uses microbial surface components recognizing adhesive matrix molecules (MSCRAMMs) to bind to the skin surface. The *S. aureus* produces a variety of cytotoxin-like hemolysins and leukocidins as well as cytolytic enzymes that cause lysis of the host tissues and help in infection. S. *aureus* also contains an array of exotoxins like staphylococcal enterotoxins A, B, and C as well as toxic shock syndrome toxin-1 [23]. *S. aureus* has various mechanisms to evade the immune cell functions and hence, to prevent bacterial clearance. There are several pore-forming toxins of *S. aureus* like α-toxin and leukotoxins, including γ-hemolysin, Panton-Valentine leukocidin (PVL), and leukocidin E/D. All these together aids this pathogen in evading the host immune system and help spreading of infections [24].

## 3. Skin Antimicrobial Peptides (AMPs)

Human AMPs are 12–100 amino acids long alpha helical molecules that are amphipathic and cationic in nature [25]. These molecules mostly fight against the pathogens by membrane interaction leading to lysis. Since, the skin forms the first line of contact with the normal microbiota and other pathogens, the skin epithelial cells produces epithelial antimicrobial proteins (eAMPs) also known as the skin AMPs, for the overall protection of the body from the various external assaults. The skin AMPs generally consists of cathelicidins and β-defensins. Several AMPs like human β-defensins (hBD) 1-3, cathelicidin LL-37, ribonuclease RNase-7, and dermcidin are found in the human skin [26]. The mechanisms of antimicrobial action of AMPs are based on either AMPs produced by skin commensal microbiota or by activation of the pathogen recognition system, which later triggers the production of AMPs from the epithelial cells [27]. The human cathelicidin antimicrobial peptide hCAP-18 is the precursor molecules for several AMPs. LL-37, a 37-amino-acid peptide molecule is the major potent AMP derived through in situ protease-digestion of hCAP-18, that lyse the negatively charged pathogen membrane by binding to it using a net positive charge of +6 [28]. The synergistic effect of AMPs produced by the commensal microbe and the skin are also documented. For example, in case of atopic dermatitis (AD) the AMPs produced by the commensals *Staphylococcus epidermidis* and *Staphylococcus hominis* were found to work synergistically with LL-37 against *Staphylococcus aureus* infection [29].

AMPs are identified to have direct impact on the immune cells like dendritic cells, macrophages, monocytes, neutrophils along with the T and B lymphocytes. Recent studies emphasise the immunomodulation properties of AMPs that can heighten the immune reaction and recognition of pathogen that evade the system. Table 1 summarizes the major types of such AMPs produced by the human skin.

### Mode of Action

Most AMPs work by direct killing via targeting the bacterial cell membrane (Figure 2). The membrane targeting includes interaction of the AMP via its binding domain to the bacterial cell membranes. Once bound, pore formation within the lipid bilayer takes place [40]. Apart from that, AMPs also cause rupture of the pathogens by targeting crucial proteins, enzymes, and cellular mechanisms [41]. The other mechanisms of their action may include the immunomodulatory ability of AMP. AMPs cause activation of the interleukins as well as chemokines and cytokines, which in turn heighten the immune response against the pathogens and lead to their clearance from the body [42].

## 4. Immunomodulation of Host Immune Cells and Responses

The concept of immunomodulation is a new-age approach of manipulating the immune response against pathogen entry and infection inside the host [44]. The immune system has an important role not only for providing defense but also helps in the wound healing process. Modulation of immune system to heighten the host protective response by training the immune cells to act against the pathogens in an efficient manner by administration of modulators is the main goal of this approach [45]. The innate immune system is the primary host defense against life threatening infections as well as host modified harmful entities that are self-harming in nature [44]. The innate immune system consists of PRRs which identifies the foreign molecules and starts producing inflammatory responses like interferons, cytokines and chemokines [46]. But with time, the pathogens have evolved ways to evade the immune system and colonize within the host and start an infection. For example, *Salmonella enterica* causes acetylation of the lipid A component of its cell membrane lipopolysaccharide, thus altering its surface charge to a positive state causing the bacterial cells to repel positively charged AMPs produced by host immune cells and hence evade the immune response [47]. So, administering immunomodulators or “Immuno-trainers” will train the immune system by providing non-specific stimulus to produce a refined response that could counter such pathogenic adaptations by various infection causing microorganisms.

### 4.1. Dendritic Cells

Dendritic cells (DCs) are the type of migratory, antigen-presenting cells derived from bone marrow that can activate and differentiate the naive T lymphocytes. DCs recognize the foreign pathogens and allergens and leads to the induction of immunogenic responses via binding to the pathogen-associated molecular patterns (PAMPs) [48]. DCs are of different subtypes depending on the phenotype and functions. The subtypes of DCs include conventionals DCs (cDCs), monocyte-derived DCs (MoDCs), plasmacytoid DCs (pDCs), interstitial DCs, dermal DCs, inflammatory DCs, Langerhans cells (LCs), and transitional DCs [49]. DCs are the key players in the recognition and subsequent packaging and presentation of the antigenic molecules to activate the downstream immune effector machinery to produce specific immune response. DCs are generally considered as the interface between the innate and adaptive immune response [50]. Upon interaction and recognition with the bacterial pathogens the DCs leads to the enhanced expression of MHC molecules. The dermal DCs causes heightened expression of the co-stimulatory receptors. CD1c^+^ DCs and the CD141^+^ DCs are the two dominant population of dermal DCs. Among which CD141^+^ DCs are important for the cross presentation of the antigenic molecules to the CD8^+^ T cells [51]. In case of bacterial infection, DCs lead to Th17 mediated response along with the activation of inflammasome receptors. Mostly the NLRP3 inflammasome is activated after priming by a PRRs followed by activation of the NF-kB, that lead to the induction of NLPR3, pro-IL-1β and pro-IL-18 and cytokines such as IL-6, IL-8 and TNF-α [52]. However, the pathogens are evolving to equip themselves with arsenals to evade DCs. For example, in case of *Staphylococcus aureus*, the DCs interact with the pathogen and cause lysis of the bacteria. Once the pathogen is engulfed by DCs, phagosome formation takes place that fuse with hydrolase containing lysosomes that cause killing of the pathogen. Pathogens are also subjected to the acidic pH and reactive oxygen and nitrogen species (ROS & RNS) [53]. However, *S. aureus* has employed several mechanisms to evade killing by the DCs. *S. aureus* produces staphyloxanthin (Sx) that neutralizes ROS and RNS. *S. aureus* also interfere with antigen processing and presentation to the MHC class II, reducing their T cell-priming ability by producing phenol-soluble modulins (PSMs) that can form pores and cause disruption of the phagolysosome, hence evade the DCs [54]. Recent studies in the field of immunomodulation show that Human beta-defensin-3 (hBD3) can induce the maturation of Langerhans cell–like dendritic cells (LC-DCs) followed by increment in the CCR7 expression. hBD3 also cause maturation of primary human skin–migratory DCs that migrate towards the draining lymph nodes for induction of immune responses. Antimicrobial Peptide (human lactoferrin-derived peptide) hLF1–11 also induces the monocyte-dendritic cell differentiation and as a result was found to enhance the Th17 polarization which shows anti-fungal effects [55]. So modulation of the DCs using AMPs can be part of the new immunotherapy regime [56].

### 4.2. Mast Cells

Mast cells are leukocytes from the hematopoietic lineage which are found in abundant amount in the host epidermal layers and blood. The mast cells contain abundant amount of secretory granules of histamine, serotonin and cytokine-like tumor necrosis factor (TNF) and IL-4, as well as growth factors, like vascular endothelial growth factor [57]. Mast cells enhance the migration of DCs into the site of infection. The mast cells are of two types based on their phenotypes—(i) mucosal mast cells that produce tryptase; and (ii) connective tissue-based mast cells that produce chymase, tryptase, and carboxypeptidases. At the site of infection and inflammation, upon activation, mast cells release their secretory granules and several chemokines and cytokines and lead to the induction of additional inflammatory mediators [58]. Mast cells are also found to produce extracellular traps known as the MC extracellular traps (MCETs) that entrap pathogens like the Group A streptococci (GAS) and kill them by action of tryptase and the antimicrobial peptide LL-37 [59]. In case of *E. coli* and *K. pneumoniae,* pathogen bind to the BMMCs by interaction of CD48, a glycosylphosphatidylinositol-mannose receptor on mast cell surface with FimH, a mannose-binding lectin present on the type-1 fimbriae [60] and hence reside inside the mast cells and evade phagocytosis. However, it was found that chemically synthesized AMPs like Retrocyclins (RC-100, RC-101) and Protegrin-1 (PG-1) caused degranulation of the human mast cells (HMCs) by Mast related G protein coupled receptor X2 (MrgX2) and act independent of the formyl peptide receptor-like 1 (FPRL1), a known receptor for AMPs [61] and hence help in efficient clearance of the pathogens.

### 4.3. Neutrophils

Neutrophils also known as the human neutrophilic polymorphonuclear leukocytes (PMNs) are integral part of the innate immune system and provide a primary line of defense against bacterial infections. They are produced in bone marrow and keep circulating in blood for hours, [62] and getting immediately recruited to the site of inflammation and infection. Neutrophils employed in the site of infection uptake the bacteria, phagocytose and kill them by ROS production or bactericidal agents. Recently, it has been found to have a mechanism known as NETosis where release of neutrophil extracellular traps (NETs) takes place [63]. The neutrophil recruitment to the site of infection is a multistep process involving detection of the pathogen by recognizing their specialized PAMPs using the host PRRs, like Toll-like receptors (TLRs) and nucleotide-binding oligomerization domain (NOD) proteins [64]. This interaction triggers the production of cytokines and chemokines like IL-8, IL-1α, IL-β, CXCL1, CXCL2, CXCL5, tumor necrosis factor (TNF). But in case of gram-positive bacteria, they have inhibitory protein that binds to the receptors and hence inhibit the ligand-receptor interaction. For example, *Staphylococcus aureus* has chemotaxis inhibitory protein of *S. aureus* (CHIPS) that inhibits the migration of the neutrophils by binding to the C5a receptor and formyl peptide receptor (FPR) [65]. In case of gram-negative bacteria, many bacteria modify their lipid A structure to evade the TLR4 recognition. They also employ masking of the epitopes that allows them to avoid recognition by PRRs [66]. Recently it has been found that innate defense regulator (IDR) can modulate the chemotactic action of the neutrophils to enhance the immune response against skin associated pathogens. KSLW (KKVVFWVKFK-NH2) is a synthetic antimicrobial peptide that showed immunomodulatory properties against the chemotactic effect on human neutrophils, increasing its migration rate in case of infection of opportunistic skin pathogens *Staphylococcus aureus* and *Pseudomonas aeruginosa* infections [67].

### 4.4. Macrophages

Macrophages are immune cells of myeloid lineage present throughout the body for immunosurveillance. They along with DCs, neutrophils and mast cells are known as ‘professional’ phagocytic cells that engulf the invading pathogens and foreign particles. Macrophages are differentiated from precursor monocytes after they extravasate from blood via endothelium [68]. There are different types of macrophages based on their anatomical site and functional characteristics at those sites, like the specialized tissue-resident macrophages which include osteoclasts present in the bone, alveolar macrophages present in the lungs, histiocytes present near the interstitial connective tissue and Kupffer cells present in the liver [69]. On the basis of functional activation, we can distinguishM1 macrophage that provides protection against several pathogens and also associated with the tumor immunity; and the M2 macrophages also known as the regulatory macrophages, having a role in wound healing and against inflammation. Tumour-associated macrophages (TAMs), as the name suggests are involved in the tumour targeting immunity [70]. In case of infection, bacteria are recognized by the epithelial cells using pathogen recognition receptors (PRRs) based on the bacteria associated molecules. Once recognized release of the pro-inflammatory cytokines and chemokines like granulocyte-macrophage colony-stimulating factor (GM-CSF), monocyte chemotactic protein-1 (MCP-1), and various interleukins (IL-6, IL-1β, and IL-8) takes place. Pathogens also contain chemoattractants that induce the macrophages to be recruited for performing phagocytosis [71]. However, certain pathogens hijack the alveolar macrophages by inhibiting the acidification of the phagosome. For example, *Mycobacterium tuberculosis* (Mtb) secretes tyrosine phosphatase PtpA, that inactivates the host vacuolar ATPase, and consequently phagosomal acidification, creating a place for the bacteria to reside and persists [72]. Aryl-based synthetic mimics of antimicrobial peptides (SMAMPs) with antimicrobial activity were found to perform immunomodulatory activities in macrophages both in the presence and in the absence of lipopolysaccharide. Hence, these synthetic peptides are considered as a promising therapy against various life threatening infections [73].

### 4.5. Gamma Delta (γδ) T Cells

Gamma delta (γδ) T cells also known as the dendritic epidermal T cells (DETCs) are a small group of CD3-positive T cells present in the peripheral blood and lymphoid tissues. They are present in close proximity of the junction between the epidermis and the dermis. γδ T cells cause induction of several cytokines, such as IL-4, IL-17, IL-21, IL-22, and IFN-γ, which are associated with various immune responses [74]. They modulate the response against skin inflammation and wound healing and form the first line of defense against pathogens. γδ T cells are categorized into three subtypes according to the expression of γ and δ chains: Vδ1 T cells, Vδ2 T cells, and Vδ3 T cells [75]. In case of bacterial infection like *Listeria monocytogenes*, the circulating γδ T cells immediately come into action which causes activation of the neutrophils and clearance of the pathogens from the system. However, the bacteria are experts in evading the immune system, and evasion from the γδ T cells is also part of it [76]. Certain pathogens evade the immune system by dysregulation of the γδ T cells. It was found that Synthetic Cationic Peptide IDR-1002 showed induction of the γδ T cells which lead to the activation of the adaptive immunity [77]. Such molecules thus could be targeted and modulated in favor of enhancing host immune responses against specific pathogens.

### 4.6. NK Cells

Natural killer (NK) cells are part of the innate immune system that has cytolytic function mostly against tumor or virus infected cells. However, recently the role of NK cells in the immune response against bacterial infections has been established. Upon activation, the NK cells induce release of cytokines like IFN-γ, tumor necrosis factor-α (TNF-α), granulocyte macrophage colony-stimulating factor (GM-CSF), and chemokines like CCL1, CCL2, CCL3, CCL4, CCL5, and CXCL8 that can regulate the role of other innate and adaptive immune cells [78]. Even though NK cells constitute a smaller fraction of the total lymphocyte population, yet they are highly widespread throughout the body. The magnitude of the NK cells-mediated cytotoxicity is dependent on the microenvironment. Type I IFN, IL-12, IL-18 and IL-15 are known activators of NK cells while IL-2 was found to promote cell proliferation, cytotoxicity and cytokine secretion in the NK cells [79]. It was found that in case of the extracellular bacterium *Staphylococcus aureus*, NK cell-depleted mice had drastically higher bacterial load in the lungs and spleen as compared to control animals [80]. In case of intracellular bacteria like *M. tuberculosis*, the NK cells caused apoptosis of infected monocytes and thus killing the bacteria. NK cells also secrete cytotoxic molecules like Granulysin, a membrane damaging peptide, which are effective against both gram-positive and -negative bacteria [81]. However, certain bacteria have evolved to utilize the NK cells to survive inside the host. For example, *Listeria monocytogenes* encodes a protein that leads to the induction of NK cells. IL-10 produced by the NK cells regress the activation of the inflammation-associated myeloid cells, resulting in an increased bacterial load inside the host [82]. Antimicrobial peptide indolicidin and its synthetic structural analogues were found to have anti-microbial activity. They were found to cause enhanced killing the bacteria through the induction of improved killing activity of NK cells [83].

### 4.7. Keratinocytes

Keratinocytes are predominantly present in all the layers of epidermis. The differentiation of the keratinocytes takes place while travelling from the outer epidermis towards the skin surface. They provide protective shield to the skin against external environment and pathogens [84]. The keratinocytes identify pathogens by the interaction of their PRRs with the PAMPs present in pathogens. Among the PRRs, TLRs) are the most common that are produced both constitutively as well as via induction, and dectin-1 and nucleotide-binding oligomerization domain (NOD)-like receptors (NLRs) also take part in the pathogen recognition [85]. The epidermal keratinocytes fight against the invading pathogens by releasing cytokines and chemokines like CXCL1, CXCL2, CCL20, CCL2 and IL-8, and also certain AMPs. Apart from that the epidermal keratinocytes are known to be non-professional phagocytic cells that internalize the invading pathogens to potentiate the clearance of those foreign material out of the body [86]. In case of *S. aureus* infection IL-17C gets activated following recognition by the NOD2 based pathway and that in turn cooperate with TNF and enhance the human β-defensin (HBD) 2 and HBD3 which cause reduction of the pathogen load. Apart from that LL-37 was also produced by the keratinocytes which is found to be effective against several pathogens [87]. But *S. aureus* was found to interact with keratinocytes PRRs and cause stimulation of the keratinocyte autophagy which due to some unknown reasons facilitates the persistence of the bacteria intracellularly by downregulating the inflammasome signaling pathway [88]. Recently, it has been found that cathelicidin peptide LL-37 if used as immunomodulatory agent causes induction of the proinflammatory cytokines in keratinocytes which enhance the bacterial clearance ability of these cells [89]. Similarly, recently the antibacterial polymers has gained popularity as an effective alternative therapy against infections for example, amino-functional hyperbranched dendritic–linear–dendritic copolymers (HBDLDs) based on polyethylene glycol (PEG) and 2,2-bis(hydroxymethyl)propionic acid (bis-MPA) are found to induce the expression of RNase 7 and hence enhance keratinocyte mediated killing of the pathogens [90].

### 4.8. Melanocytes

Melanocytes are mainly known as the melanin pigment producing cells which are responsible for the protection against UV-induced DNA damage. The melanocytes are the dendritic cells that are mostly present in the epidermis and in the hair follicles [91]. Melanocytes are derived from the neural-crest, and they migrate while embryological development takes place and localize in the skin and hair follicles [92]. The immunological role of the melanocytes in the invertebrates is well studied and validated. However, in case of the human melanocytes it has been found that in case of pathogens like *Candida albicans* the TLR2 and 4 expression heightens in the melanocytes which inhibit pathogen growth [93]. As already known that TLR2 is involved in the recognition of gram-positive bacteria based on the peptidoglycan and lipoteichoic acid, and TLR4 recognizes the peptidoglycan present in the gram-negative bacteria cell surface. Once recognized myeloid differentiation factor-88 (MyD88) gets recruited, which allows activation of the nuclear factor-κB (NF-κB) pathway or mitogen-activated protein (MAP) kinase pathway that leads to the pathogen clearance from the system [94]. Recently, it was found that pathogens like *Salmonella typhimurium* can reside inside the melanocytes and hence evade the host defense system [95]. However, alpha-melanocyte stimulating hormone (α-MSH) is a newly discovered neuropeptide from the melanocortin family, which has been found to enhance the activity of the melanocytes leading to enhanced antimicrobial activity and bacterial clearance [96].

## 5. AMPs in Adjunct Therapy

Recently, several AMPs have undergone various stages of preclinical and clinical phase trials to combat skin related bacterial infections. AMPs has recently gained highlight as promising alternatives to combat bacterial infections and control microbial resistance [97]. The main steps in the AMP therapeutic development must include: (i) to screen for AMPs that are effective against prevalent multi-drug resistant opportunistic pathogens; (ii) to evaluate the impact of immunomodulation by AMPs in treating skin infections; (iii) to identify new techniques for better delivery of the AMPs. Some of the examples of AMPs in clinical trials as therapeutics include, IDR-1018 already in clinical trials (phase II) for healing wounds [98]. AG-30, an angiogenic peptide that showed high antimicrobial activity against *P. aeruginosa*, *E. coli*, and *S. aureus* via a membrane disruption mechanism [99]. Histatin 5 (Hst 5) is a salivary cationic peptide produced with anti-bacterial activity against five out of six ESKAPE pathogens (*Enterococcus faecium*, *Staphylococcus aureus*, *Klebsiella pneumoniae*, Acinetobacter *baumanni*, *Pseudomonas aeruginosa*, and *Enterobacter* species) pathogens [32]. AMPs are also used as adjuvants for the development of vaccines against viral and bacterial infections. hBD-2 is being used as an adjuvant to the vaccine against Mycobacteroides, has increased the effectiveness of therapy [100]. Pexiganan, a derivative of magainin, is under phase III trials for the treatment of infected foot ulcers in patients with diabetes mellitus [34]. Synthetic peptides such as P-novispirin G10 which has a bactericidal immunomodulatory fusion peptide of HBD-3 with a mannose-binding lectin, showed high effectivity against MRSA [101].

### Challenges of AMP Therapies

Several AMPs are currently undergoing clinical trial for replacing the conventional antibiotics to treat several bacterial infections. However, one of the major problems associated with AMPs is their cytotoxic nature [102]. Because of the possible cytotoxicity of AMPs the clinical trial of many AMPs are not possible. For example, tyrothricin can only be administered topically because of its systemic toxicity. Recently, there are strategies like developing synthetic analogues of AMPs that are less toxic in nature for example, truncated LL-37 fragments, named as LL-13 and LL-17, are less toxic in nature and are highly effective against both MRSA and VRSA *Staphylococcus aureus* strains [103]. Apart from cytotoxicity another problem with AMP therapy is the delivery of AMPs to the site of infection. To solve this problem several novel AMP delivery systems are being developed. For example, nanotubes attached AMPs like graphene oxide nanotubes attached AMPs are found highly effective against *Staphylococcus aureus* (MRSA) strains [104]. Hydrogels loaded with alamethicin are found to be effective in reducing bacterial adherence and fastening the wound healing [105].

## 6. Conclusions

Identification and development of new therapeutic strategies to deal with infections caused by antibiotic-resistant bacteria continues to be one of the major problems in the field of drug discovery. AMPs are found to be highly effective against multi-drug resistant opportunistic skin pathogens. Apart from that, the concept of immunomodulation has opened a new avenue in the field of therapeutics. AMPs can be exploited to modulate not only the innate immune responses but also to deal with infections caused by antibiotic-resistant bacteria. It will aid not only to fight against skin pathogens but also to treat inflammatory skin diseases and promote wound healing. Recent advances in the understanding of the cellular and molecular functions and mechanisms of AMPs in human skin and in infectious/inflammatory skin diseases, will contribute to have better target medicines and therapies.

## Figures and Tables

**Figure 1 antibiotics-13-00439-f001:**
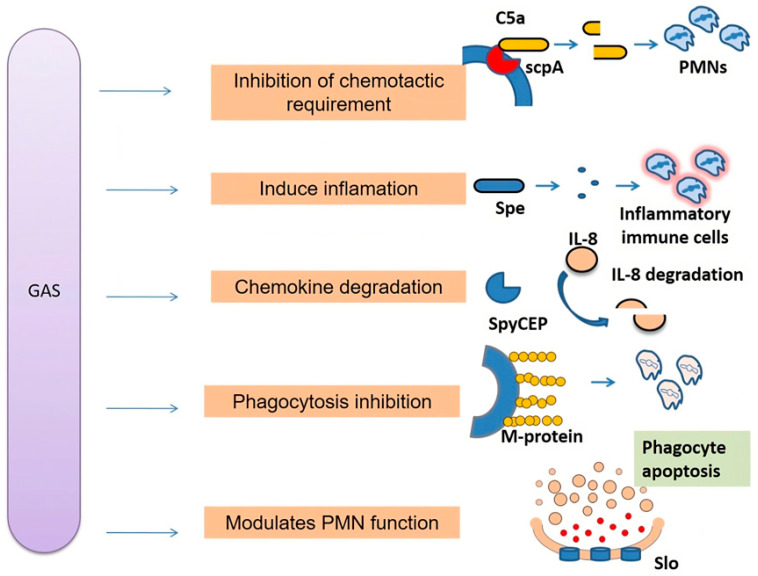
Pathogenesis of GAS: Group A streptococcus (GAS) invade immune system by employing different virulent factors, such as scpA which cleaves C5a and spyCEP which targets IL8, and affect recruitment of immune cells. M-protein which provides protection against phagocytosis and SLO impairs neutrophil function [20].

**Figure 2 antibiotics-13-00439-f002:**
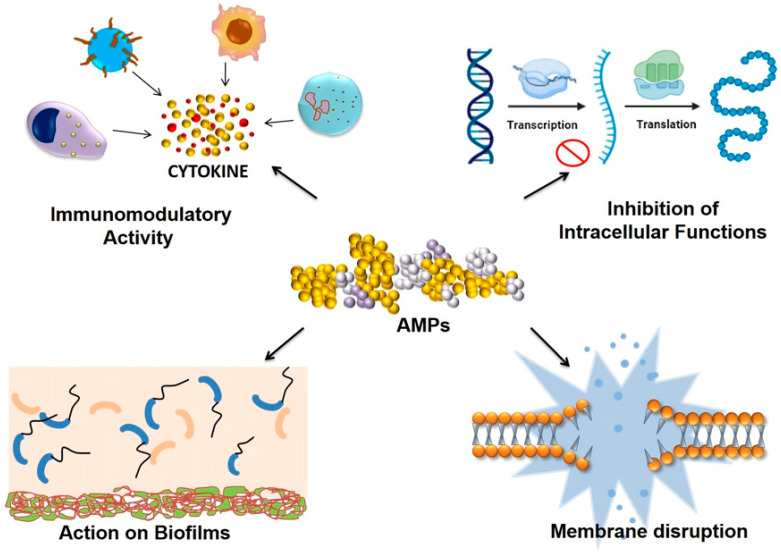
Role of antimicrobial peptide: regulating immunomodulatory activity of immune cells, inhibition of intracellular function of cells, direct killing of bacteria by disruption of membrane, and acting on biofilm produced by bacteria [43].

**Table 1 antibiotics-13-00439-t001:** Different AMPs produced by the human skin.

Sl. No.	Antimicrobial Peptide	Mechanism of Action	Reference
1	LL37	Barrel-stave mechanism of membrane disruption and inhibit LPS binding in Bacteria, fungi and viral pathogens; *P. aeruginosa*	[30]
2	OP-145 (LL-37 derived; phase II)	Membrane disruption in gram-positive	[31]
3	PAC113 (P113; histatin 5 analog; phase IIb)	Membrane disruption and immunomodulation ESKAPE Pathogens	[32]
4	Cys-KR12	Membrane disruption *E. coli*, *S. aureus*	[33]
5	LTX-109	Membrane disruption and cell lysis in MRSA	[34]
6	AMPR-11	Disrupts bacterial membranes by interacting with cardiolipin and lipid A in sepsis-causing bacteria, including multidrug-resistant strains	[35]
7	Dalbavancin	Inhibition of bacterial cell wall synthesis in *S. aureus*	[36]
8	Polymyxins	Membrane disruption in *P. aeruginosa*	[37]
9	Vancomycin	Inhibition of bacterial cell wall synthesis in MRSA, VISA, VRSA	[38]
10	WRL3	Membrane lysis in MRSA	[39]

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
