# Peer review of "AMPs as Host-Directed Immunomodulatory Agents against Skin Infections Caused by Opportunistic Bacterial Pathogens"

_antibiotics, 2024, doi:10.3390/antibiotics13050439_

Round 1

Reviewer 1 Report

Comments and Suggestions for Authors

This review is about the role of antimicrobial peptides against skin infections. The authors not only consider the well-known direct action of AMPs on pathogen microorganisms but also they remark the promising use of AMPs at sub-antibacterial concentrations as modulators of the immune response. The utilization of AMPs as immunomodulators to enhance immune response against pathogens paves the way for alternative therapies that could circumvent drug-resitance issues. The authors explain the immunomodulatory functoins of AMPs at different levels, explaining the role of the different components of the immune system and how AMPs can act on them in order to achieve therapeutic effects.

In general, this is an interesting manuscript with potential interest for the field of peptide-based drug discovery which is usually oriented to the direct antimicrobial power but not to the immunomodulatory properties. I think that the bibliographical compilation is correct, but I would like to point some issues that have to be addressed:

A. I suggest a new section organization to improve the information flow and get a more coherent outline of the manuscript:

1. Introduction

2. Opportunistic skin pathogen-associated infections

2.1 Group A streptococcal infections

2.2 Staphylococcal infections

3. Human antimicrobial peptides (AMPs)

3.1 Mode of action of AMPs

4. Immunomodulation of host immune cells and responses

4.1 Dendritic cells

4.2 Mast cells

4.3 Neutrophils

4.4 Macrophages

4.5 Gamma delta T cells

4.6 NK cells

4.6 Keratinocytes

4.7 Melanocytes

5. AMPs in adjunct therapy

5.1 Challenges of AMP therapies

6. Conclusion

B. Throughout the text there are many sentences in which a term is repeated several times, which makes them sound repetitive. Please, try to find synonyms or rewrite the sentences. For example:

Lines 103-105: [...] life threatening INFECTIONS. Group A streptococcus causes INFECTIONS ranging from mild INFECTIONS [...]. 

Lines 297: [...] from precursos MONOCYTES, afther the MONOCYTES extravasate [...]

C. In some places, there are long, complex word constructions that should be rewriten in order to facilitate the reading. For example:

Lines 204-205: Dendritic cells are the type of migratory bone marrow derived antigen-presenting cells [...] 

I suggest to change it for "Dendritic cells are a type of migratory, antigen-presenting cells derived from bone marrow"

D. There are a lot of abbreviations in this text and I strongly recommend that the first time they appear you should define it, for example: Antimicrobial peptides (AMPs). Once you have defined it, please use only the abbrevation in the rest of the text (in this case, after the first definition, use always AMPs to refer to antimicrobial peptides). This is the usual way to do it in scientific publications, and instead of that, you use abbrevations and complete definitions equally throughout the text -you should avoid this-. 

E. All the figures lack of their captions. Please, add an appropriate caption for every figure of the manuscript.

F. In my opinion, one interesting thing of this review is to find a summary of AMPs that have demonstrate their immunomodulatory properties. You mention some of them in sections 6 to 13. It would be very useful for the reader to add a table summarizing the immunomodulatory AMPs, including which immune cell type they act on, the corresponding reference, whether they are on clinical trials...

G. Finally, I have the next some minor details:

Line 53: I would add "From a cellular point of view", the skin consists of...

Line 57: It should say "express" instead of "expresses"

Line 57: The abbreviation for pathogen-pattern recognition receptors is PRRs, not PARPs.

Line 69: The word "for" has a different letter size (bigger).

Line 74: Please change "microflora" for the more accurate term "microbiota".

Line 75: Please change "microflora" for the more accurate term "microbiota".

Line 77: Remove the period after "body".

Line 80: Remove the comma after "Staphopain A".

Line 82: I would write "group A streptococcus (GAS) produces..."

Line 82: Pneumolysin is produced by Streptococcus pneumoniae which doesn't belong to group A          streptococcus.

Line 88: It shoulds say "AMPs" instead of "AMPS".

Line 90: Remove "antimicrobial peptides" and use only the abbreviation AMPs.

Line 94: Remove "antimicrobial peptides" and use only the abbreviation AMPs.

Line 101: It should say "allow" instead of "allows".

Line 101: It should say "gram-positive" instead of "gram positive".

Line 105: Change one of the "infections" for other word to avoid repetitions ("condition", "case",               "disease", "illness").

Line 113: Remove the word "in" after colonize.

Line 114: Add a comma after "Apart from that".

Line 119: The open parenthesis is in a different font size.

Line 119: Please write (TNFβ) after tumor necrosis factor beta, and add (IFNγ) after gamma interferon.

Line 124: Please change "flora" for the more accurate term "microbiota".

Line 125: Change "compromisation" (it doesn't exist) for "compromise".

Line 126: Please change one of the "infectoins" for other synonym.

Line 127: Please define the abbreviation "TSS".

Line 132: It should say "cytotoxin-like" instead of "cytotoxin like".

Line 133: Please change the second "and" for "as well as", to avoid repetition.

Line 139: Change "S. aureus" for "this pathogen" to avoid repetition.

Line 141: I would change the section title for "Skin antimicrobial peptides (AMPs)".

Line 142: Remove "antimicrobial peptides" and use only the abbreviation AMPs.

Line 146: Remove "antimicrobial peptides" and use only the abbreviation AMPs.

Line 155: It should say "lyses" instead of "lyse".

Line 156: Change the second "mebrane" for "it" to avoid repetition.

Line 164: Change "property" for "properties".

Line 165: Change "summarized" for "summarizes".

Table 1: First column title: what does it mean "SI No."?; remove the blank space before "inhibit" in the          LL37 row; in the OP-145 row change "gram positive" for "gram-positive"; in the LTX-109 row,          change Methicillin-resistant staphylococcus aureus for "MRSA", as it has been defined before. 

Line 170: Change "Mostly" for "Most".

Line 171: Change the first "binding" for "interaction", to avoid repetition.

Line 173: Add a comma after "Apart from that".

Line 192: Remove "patter recognition receptors" and use only the abbreviation PRRs, as it has been               defined before.

Line 194: Add a comma after "But with the time".

Line 204: Change the sentence construction for "Dendritic cells are a type of migratory, antigen-           presenting cells derived from bone marrow".

Line 222: Remove "patter recognition receptors" and use only the abbreviation PRRs, as it has been               defined before.

Line 227: It should say "fuses" instead of "fuse".

Line 228: Change "that cause killing" for "causing the killing".

Line 237: It should say "causes" instead of "cause".

Line 246: It should say "cytokine-like" instead of "cytokines like".

Line 247: Add commas: "IL4, as well as growth factors, like"

Line 248: Something is missing as it say "ells".

Line 249: Rewrite: "The mast cells are of two types on the basis of their phenotypes: i. mucosal mast   cells that produce tryptase; and ii. connective tissue-based mast cells that produce   chymase.." 

Line 251: Change "on activation" for "upon activation".

Line 252: Change "as well as" for "and".

Line 254: The word "trap" is repeated three times in this sentence, please find synonyms.

Line 258: Change "to FimH" for "with FimH".

Line 267: Change "HUMAN" for "human".

Line 270: Change "for hours and get immediately" for "for hours, getting immediately"

Line 273: Adde a comma after "Recently".

Line 274: It should say "takes place" instead of "take place".

Line 277: Remove "patter recognition receptors" and use only the abbreviation PRRs, as it has been               defined before.

Line 281: It should say "ligand-receptor interaction" instead of "ligand-receptorinteraction".

Line 282: It should say "inhibits" instead of "inhibit".

Line 291: Rewrite the whole sentence as "increasing its migration rate in case of infections of    opportunistic skin pathogens".

Line 295: Change "mast cells are called as the professional" for "mast cells are known as professional"

Line 297: Change the second "monocytes" for "they" to avoid repetition.

Line 300: Please separate the elements of the enumartion with semicolons (;).

Line 302: It should say "On the basis" instead of "On basis".

Line 303: Change the sentence for "On the basis of functional activfation, we can distinguish M1   macrophages.."

Line 306: Add commas "TAMs, as the name suggest, are involved".

Line 307: Remove "the" from "the bacteria".

Line 315: Add the complete name of the pathogen "Mycobaterium tuberculosis".

Line 315: Add a comma after PtpA.

Line 320: Add a comma after "Hence".

Line 320: It should say "are considered as a promising".

Line 323: It should say "are a small group".

Line 327: Add a comma after IFNγ.

Line 328: Remove the comma after inflammation.

Line 332: It should say "into action" instead of "in to action".

Line 343: Add a comma after "Upon activation"

Line 344: Use only the abbreviation for IFNγ.

Line 355: It should say "killing the bacteria" instead of "killing of the bacteria".

Line 355: Add an "a" before "membrane damaging peptide".

Line 368: It should say "differentation of the keratinocytes" instead of "differentiation in the   keratinocytes". 

Lines 371-373: Please use only the abbreviations of PRRs, PAMPs and TLRs, as they were defined before.

Line 399: It should say "that are mostly" instead of "that mostly".

Line 409: add a comma after "gets recruited".

Line 418: Change the sentence to "several AMPs have undergone various stages".

Line 456: Rewrite as "to deal with infections caused by antibiotic-resistant bacteria".

Line 459: It should say "Apart from that, the concept of".

Comments on the Quality of English Language

Writing of the manuscript should be improved. The two major issues regarding English quality are the presence of some intricate sentences that should be simplified and the iteration of words in the same sentences. Plase, try to simplify long, complex sentences and find synonims to avoid excessive repetition of some words. 

Reviewer 2 Report

Comments and Suggestions for Authors

-The review is quite interesting and complete, but the authors need to review it carefully

-The references are terribly organized. The authors did not follow the instructions for writing the article. Please read Instructions for Authors

References: References must be numbered in order of appearance in the text (including table captions and figure legends) and listed individually at the end of the manuscript.

-Line 69.- Please check font size

-Line 248.- Please check redaction. Mast cells enhances the migration of DCs and ells in to the site of infection.

-Line 267.- HUMAN in lower case

-What does mean Sl.No. and MOA in the table 1

-The figure 2 doesn´t have figure legend

Round 2

Reviewer 2 Report

Comments and Suggestions for Authors

Dear authors, I would have liked to see the final version of the article without the deletions. I consider that the writing has been improved. However, citation 51 is incomplete, please correct it.

 51.- Nesmiyanov, P. P. (2020). Dendritic cells.